# Intrapartum Analgesia—Have Women’s Preferences Changed over the Last Decade?

**DOI:** 10.3390/medicina58010087

**Published:** 2022-01-07

**Authors:** Agnieszka Jodzis, Maciej Walędziak, Krzysztof Czajkowski, Anna Różańska-Walędziak

**Affiliations:** 12nd Department of Obstetrics and Gynecology, Medical University of Warsaw, Karowa 2 St., 00-315 Warsaw, Poland; agnieszkajodzis@gmail.com (A.J.); krzysztof.czajkowski@wum.edu.pl (K.C.); aniaroza@tlen.pl (A.R.-W.); 2Department of General, Oncological, Metabolic and Thoracic Surgery, Military Institute of Medicine, Szaserów 128 St., 04-141 Warsaw, Poland

**Keywords:** vaginal delivery, epidural analgesia, pain relief in labor, maternal preference, shared decision making

## Abstract

Background: Global access to social media has supposedly changed women’s awareness about the pharmacological and alternative methods of pain relief during vaginal delivery. The purpose of the study was to analyze changes in women’s preference and opinion about different forms of labor analgesia over the past decade. Materials and methods: The study was designed as an anonymous survey with questions about women’s knowledge and preference of different forms of pain relief in labor. The survey was conducted in 2010 and 2020, with data collected from 1175 women in 2010 and 1033 in 2020. Results: There were no differences between 2010 and 2020 in the proportion of women who wanted to receive analgesia in labor, at, respectively 67.9% of women in 2010 and 73.9% in 2020. About 50% of women chose epidural analgesia as the only efficacious method of pain relief in labor both in 2010 and 2020. There were no differences between the two time-points in the distribution of chosen methods of pain relief. In total, 92.3% of women in 2010 and 94.9% in 2020 thought that they should have the possibility of independent choice of analgesia method before the delivery (*p* < 0.04). Conclusions: A high proportion of Polish women choose EDA over other pharmacological and nonpharmacological methods of pain relief in labor, and this preference has not changed over the last decade. Increasing women’s knowledge about different methods of intrapartum pain relief may lead to wider use of nonpharmacological methods of pain relief.

## 1. Introduction

The fear of labor pain is one of the main reasons for women’s preference for cesarean section (CS) over vaginal delivery, one of the most important factors responsible for the global increase in the proportion of cesarean deliveries. Increasing women’s awareness of different possibilities of pain relief in labor may influence their choice of way of delivery. Even with analgesia, there are many factors influencing maternal satisfaction with intrapartum pain control, including the intensity of pain during the first and second stage of labor, delay in providing epidural analgesia (EDA) or postpartum headache [1]. However, administration of EDA before adequate cervical dilation may be associated with longer labor [2], and longer labor is associated with a significantly lower level of satisfaction with birth experience [3].

Global access to social media and high popularity of the subject of labor have changed women’s awareness about the pharmacological and alternative methods of pain relief during vaginal delivery. Women’s expectations are influenced by their background; they obtain information more voluntarily from their family, friends or the internet rather than from healthcare professionals [4]. The World Health Organization (WHO) guidelines from 2018 stress the importance of women-centered care based on individual needs, with the aim of a positive birth experience [5]. Although pharmacological methods of pain relief are associated with more efficient pain relief than nonpharmacological ones, some women prefer active work during childbirth, finding it more gratifying [6,7]. Anxiety, depression and fear of childbirth increase the preference for EDA [8]. Women discuss their preferences with their obstetricians and decide about the preferred method of labor analgesia many weeks before the expected date of delivery and prepare their birth plans, including pain management during labor [4]. The preferences also differ between obstetricians, who tend to choose pharmacological methods, and midwives, who often prefer nonpharmacological methods. Women need to be provided with adequate information by the healthcare professionals, not influenced by their personal preferences [6]. Although primiparas compared to multiparous women are more likely to choose nonpharmacological methods in their birth plans, their decision often changes after the onset of labor contractions and finally they use more pharmacological pain relief than intended [9,10]. The purpose of the study was to analyze changes in women’s preference and opinion about different forms of labor analgesia over the past decade.

## 2. Materials and Methods

This study was designed as an anonymous survey with questions about women’s awareness and preference of different forms of pain relief in labor. The questionnaire included the possible options of EDA, pethidine injection, pudendal blockade, delivery in water, warm bath, breathing techniques, massage, music therapy, mental training, aromatherapy, acupressure and acupuncture. Additionally, the questionnaire included questions about women’s knowledge and preference of mode of delivery, which are presented in our other studies. The questionnaire was distributed as a paper survey and via social media. We conducted the same survey in 2010 and in 2020 and collected data from 1175 women in 2010 and 1033 in 2020. The exclusion criteria were not female gender, minority and missing or conflicting data.

### 2.1. Statistical Analysis

Statistical analysis was performed using Statistica 13 (StatSoft. Inc., Tulsa, OK, USA). U-Mann–Whitney test and Student’s t-tests were used for quantitative data comparison as required. Two-sided Fisher’s exact test and chi-squared test were used for categorical and binary data comparison as required. A *p* value < 0.05 was considered significant.

### 2.2. Ethical Considerations

The study was anonymous, performed in accordance with the ethical standards laid down in the 1964 Declaration of Helsinki and its latter amendments (Fortaleza). All participants were informed about the purpose of the study and informed consent was obtained electronically prior to the beginning of the survey. The approval from Warsaw Medical University Ethics Committee was obtained on 19-03-2013, with code AKBE/21/13.3.

## 3. Results

The majority of patients that filled in the questionnaire at both time-points were of reproductive age, 95% in 2010 and 98% in 2020, with the medium age of 28.8 years in 2010 (SD 8.8) and 32.0 in 2020 (SD 6.7). The vast majority of respondents in both groups had medium or high socioeconomic status (93.7% in 2010 vs. 98.6% in 2020), and most of them lived in cities of more than 50,000 habitants (68% at both time-points). Overall, 49.5% of women in 2010 and 63.7% in 2020 had higher education. In total, 45% of 2010 respondents had history of at least one delivery, compared to 78.4% of 2020 respondents.

The baseline characteristics of the groups are presented in Table 1.

Overall, 90.0% of women in 2010 and 86.7% in 2020 considered delivery as a painful event, with only 4.6% in 2010 and 3.0% in 2020 who thought it was not painful. Only 36.2% of respondents in 2010 and 51.7% in 2020 estimated that cesarean delivery was painful. In 2010, 40.0% of respondents and 30.0% in 2020 who considered it as not painful.

There were no differences between 2010 and 2020 in the proportion of women who wanted to have analgesia in labor, with 67.9% of women in 2010 and 73.9% in 2020. In total, 16.0% of women in 2010 and 12.1% in 2020 declared they did not want analgesia during labor. Further, 16.2% of respondents in 2010 and 14.0% in 2020 had no preference. The respondents had a possibility of choosing a random number of analgesia methods they considered effective in intrapartum pain reduction. Half of the 2010 and 2020 groups chose EDA as the only efficacious method. Among all respondents, both in 2010 and 2020, 64.5% indicated EDA as an efficient way of relieving pain in labor, 46.3%—delivery in water, 46.2%—warm bath, 46.1%—mental training, 38.4%—massage, 13.4%—music therapy; all other methods were chosen by less than 10% of respondents. There were no differences between the 2010 and 2020 groups in the distribution of chosen methods of pain relief. 

The vast majority of women both in 2010 and 2020 thought that they should have the possibility of independent choice of analgesia method before the delivery, at 92.3% of women in 2010 vs. 94.9% in 2020 (*p* < 0.05). Additionally, 2.7% of respondents were against this in 2010 and 1.8% in 2020. Overall, 5.0% in 2010 and 3.7% in 2020 had no preference. 

Women were also asked about the moment when the patient should be given analgesia, and 40.2% of respondents in 2010 and 39.3% in 2020 thought that analgesia should be given at the moment chosen by the patient, 9.8% vs. 12.8%—on the onset of pain, 17.4% vs. 14.5%—at 3 cm of dilatation, 5.3% vs. 5.6%—on the onset of expulsive pains, 3.7% vs. 4.3%—at 8 cm dilatation and 1.7% vs. 1.2%—at hospital admission. There were no statistically significant differences found among the distribution of the responses in 2010 and 2020. Only less than a half of respondents at both time-points were aware that there were possible complications of analgesia. Among those who recognized the possibility of complications, 38.1% indicated headache, 22.5%—paresis, 22.5%—paralysis of lower limbs and 11.1%—deterioration of newborn’s wellbeing. Results are presented in Table 2.

## 4. Discussion

We analyzed women’s preference and knowledge about different methods of intrapartum analgesia in order to verify if there were any differences over the last decade. We found that there were no changes in the general proportion of women who wanted to have pain relief in labor, as well as in the proportion of those who wanted to have EDA, which was around 50% of the respondents both in 2010 and 2020. The number of women who were aware of the possible complications of intrapartum analgesia did not change either, nor did the proportion of those who wanted to choose the moment of administration of analgesia, which remained at the level of 40% at both time-points. Improving women’s knowledge about different methods of intrapartum pain relief would be beneficial for reducing anxiety levels and amelioration of the birth experience according to WHO standards [5].

In a Swedish study from 2019 by Westergren et al., only 19.6% of multiparas and 16.7% primiparas indicated EDA as their first choice of pain relief in labor, and 44.2% of the general group considered EDA as their second choice or last resort [9]. In our study, EDA was the first choice for 50% of women. In the Swedish study, 69.2% of primiparas and 39.2% of multiparas wanted to decide about adequate methods of pain relief after having conferred with the midwife, in the spur of the moment, compared to 40% in our study. There were differences between the groups, as the Swedish group included 239 pregnant women, and in our study, 18.3% of 1175 respondents in 2010 and 9.0% of 1033 in 2020 were pregnant.

An earlier Swedish study from 2015, by Lindholm et al. presented analgesia preference of 936 pregnant women. Overall, 79% of women preferred nitrous oxide, 63%—bath, 44%—massage, 37%—EDA, 28%—breathing techniques. Some methods were less popular—19% women preferred mental training, 14%—acupuncture, 6%—pudendal blockade and only 4% pethidine. Other methods were TENS (7%) and sterile water injections (2%). Nitrous oxide was not included in our questionnaire due to lack of its availability in our center in 2010; we decided not to include it in the 2020 questionnaire either, as we wanted the questionnaire to remain unchanged at both our time-points. In our study, 64.5% of women indicated EDA as the most efficient method of pain relief (more than three times more than in the Swedish group), followed by warm bath—46.3%, breathing techniques—46.1%, massage—38.4%, music therapy −13.4%. Less than 10% of women chose other methods, among whom 7.4% indicated pethidine (almost two times more than on the Swedish study). A comparison of women’s preferences between different studies is presented in Figure 1.

Our results show a tendency for choosing pharmacological methods of intrapartum analgesia in Polish women, a trend that has not changed over the last decade. Our observations suggest there might be lack of proper counseling about the possible methods of pain relief during labor given by the healthcare professionals. Women have insufficient information about nonpharmacological methods of pain relief. One of the reasons might be a different model of obstetric care between countries, as in our country care in pregnancy is provided mostly by obstetricians who have preference for pharmacological methods, and the role of midwives, who are observed to prefer nonpharmacological methods, is reduced to taking care of women only during prenatal classes and labor itself [6]. Swedish public policies make women more secure in labor and more aware of labor pain than do Polish public policies, as there is no sufficient and easily accessible information about intrapartum analgesia provided in Poland. The main source of professional information are the obstetricians, who often do not have time to thoroughly provide such information, as the time dedicated for one pregnancy appointment in the public health service is very limited, and the public health service is the main line of pregnancy care in Poland. Birth plans with the choice of preferred method of analgesia are introduced at labor preparation classes, but their accessibility and attendance are still limited. The rate of pregnancies taken care of by midwives instead of obstetricians is still negligible in Poland, whereas their role is of crucial importance in Sweden and other northern European countries. In many cases, information about possible methods of intrapartum analgesia is given exclusively at the beginning of labor. 

A Norwegian study from 2017 analyzed the characteristics of a group of 540 women who indicated at the 32nd week of pregnancy EDA during labor as their method of choice. The primary group included 2596 women, 21% of whom stated they would choose EDA. Among factors influencing their choice, a consultation for pregnancy concern was found to be highly associated with preference for EDA. Conversely, participation in labor preparation courses was significantly associated with a reduction in intention to use EDA during labor. [11] The results of the Norwegian study led to a thought that increasing the popularity and stressing the importance of labor preparation classes might reduce the number of Polish women who indicate EDA as the best and preferable option for pain relief during labor. We can take conclusions from the Norwegian study, however the participating populations cannot be compared, as only a proportion of our group was pregnant at the time of filling in the survey. Additionally, the level of awareness of Norwegian and Polish women is incomparable, because there is no adequate medical information about possible methods of labor analgesia provided in our country. 

In our study we included only women capable of computer and internet use, as the survey was filled in by the means of internet, therefore we might have excluded some of the women from the lowest income groups or with disabilities. Additionally, there was a difference between 2010 and 2020 groups as EDA was available free of charge to all women in 2020, only limited by the availability of an anesthesiologist. In 2010, EDA was free of charge only in case of medical indications, and there was a medium-level fee to be paid by women without indications. The acceptance of the fee might have been influenced by the medium or high socioeconomic status declared by most of our participants, and therefore influenced our results. 

The choice of EDA can be influenced by different factors, including cultural beliefs—as labor pain is considered a virtue in Japan, the rate of analgesic delivery in Japan is very low, at the level of 6.2% in 2016. The low rate of labor analgesia in Japan is also influenced by the difficulties in access to birth facilities and shortage of anesthesiologists [12]. Shishido et al. found Decision Aids (algorithms prepared to facilitate the decision about EDA in labor) useful in assisting informed decision making, and found a significant increase in patient’s satisfaction with their choice when compared to a group of women who received only informative pamphlets [13]. An antenatal anesthetic consultation also helps in lowering the anxiety level, helps in decision making and is advised especially for obese women [14]. However, any antenatal consultation about the possibilities of labor analgesia provided by a health professional is beneficial for the pregnant women’s decision making during delivery [15,16].

There is contradictory information about the influence of parity on the primary choice of EDA, with some stating that primiparas less often than multiparas include EDA in their labor plans [9,17], with others on the contrary indicating that primiparas choose to have EDA more often than multiparas [18]. An Australian study on a group of 2445 women indicated that women who used EDA in previous labors were more likely to have EDA, as well as those with higher education or of higher income. Preference for EDA of their partner or previous good experience with EDA of a family member also encouraged women to choose EDA. The study also showed that a history of previous CS increased likelihood of using EDA (OR = 13.3), as well as an instrumental delivery (OR = 2.21) [18]. Additionally, EDA during a trial of labor after CS can significantly increase a chance for successful vaginal birth [17]. Another Australian study by Steel et al., on a group of 1835 women, stated that women who used complementary and alternative medicine techniques—acupuncture, yoga classes, herbal medicine, aromatherapy oils, etc. in pregnancy (almost 50%), were significantly more likely to use nonpharmacological methods of pain relief in labor [19]. In a recent French study, multiparity and spontaneous dilatation of more than 5 cm without oxytocin use were found to be positively associated with the choice of nonpharmacological methods of pain relief in labor [20].

Alshahrani analyzed awareness of methods of in relief in a group of 416 pregnant women in Saudi Arabia. They found that 58.7% of women were unaware of any possibilities of labor pain relief, and 79.8% were unaware of the different forms of labor pain relief available. In total, 72% of women believed that spousal and family support would relief the labor pain. Further, 60% of the study group were interested in different forms of pain relief, although 86% of participants believed that pain relief would have a negative effect on mother and baby [21]. In another study from Saudi Arabia, Alshabibi et al. presented methods of intrapartum analgesia chosen by 1550 women who had vaginal deliveries at their center. Overall, 34.8% of women had intramuscular analgesia, followed by 31.8% who had EDA, and 0.6%—spinal analgesia. A total of 31.9% of women used no methods of pain relief [22]. In our study, 70.8% of participants were interested in intrapartum analgesia and 14.5% had no opinion about different types of analgesia. 

Pain relief in labor in low-resource settings is often neglected. Not only the accessibility, but also the level of knowledge and awareness of possible methods of intrapartum analgesia in low-income countries are also incomparably lower than in developed countries. In a study by Ogboli-Nwasor et al., [23] pregnant Nigerian women were asked about their awareness of methods of pain relief in labor. Their mean gestational age was 31.5 weeks, and parity of two, and 35.5% of participants were primiparous. In total, 87.3% of women had heard about pain relief methods, with the hospital being the source in 79% of cases. Additionally, 85.1% of respondents admitted that they had been counseled to use pain relief agents during their labor. Overall, 45.2% consented to the use of pain relief in labor in their current pregnancies, 92.9% of whom preferred epidural analgesia. The most common method the participants had heard about was EDA—69.4%. Only 4% of respondents remembered ever using any form of pain relief agent in labor, of which three received parenteral opioids. Another Nigerian study by Okojie et al. [24] presented an even lower level of awareness about intrapartum analgesia among Nigerian women. Of 405 women were included in the study, 79.5% were not aware of epidural analgesia. Of the 83 participants who knew about the existence of EDA, 22.9% knew it was used to relieve labor pain and 24.1% were informed by the health professionals. However, the majority of the respondents—76.5%—would accept EDA if it was offered to them in labor. 

### Limitations of Study

The most important limitations of our study are the possible recall bias and subjectivity of women’s responses and opinions. Additionally, the questionnaire was filled in by women capable of computer and internet use, therefore we might have excluded some of the women from the lowest income groups or with disabilities. There was also a change in accessibility of EDA during labor between the time-points of conducting survey, as in 2010 in our country it was free of charge only for women with medical indications and in 2020 it was available for all women without contraindications. We did not add nitrous oxide to the 2020 survey, as it was not available in our obstetric center in 2010, therefore our analysis did not include women’s opinions about the use of nitrous oxide. We also are not able to calculate the response rate, as the survey was mostly distributed via social media. There was no incentive to introduce dishonesty into responses.

## 5. Conclusions

A high proportion of Polish women choose EDA over other pharmacological and nonpharmacological methods of pain relief in labor. This preference has not changed over the last decade, which stands against the global trend to reduce labor medicalization and improve birth experience. This alarming fact introduces the question about lack of adequate information presented to women by their obstetricians and insufficient accessibility and attendance at labor preparation courses, which remain two sources of medical information about possibilities of intrapartum analgesia in Poland. Birth plans that include and explain possible methods of intrapartum analgesia should be widely distributed by health providers and available to all pregnant women. Increasing women’s knowledge about different methods of intrapartum pain relief may lead to wider use of nonpharmacological methods of pain relief and general improvements in women’s birth experience.

## Figures and Tables

**Figure 1 medicina-58-00087-f001:**
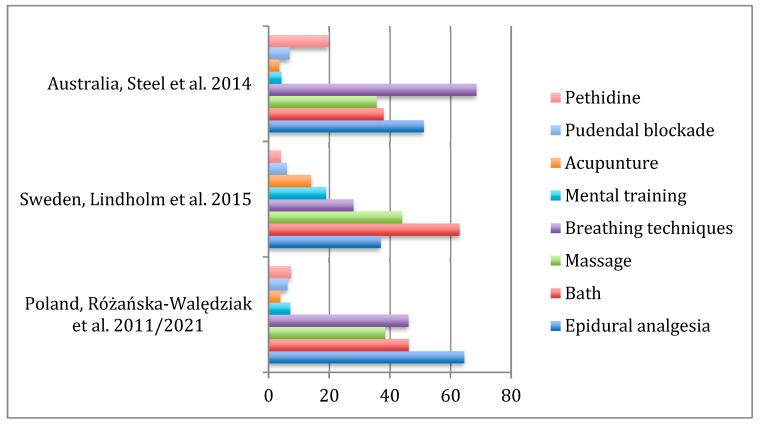
Distribution of preference for pain relief methods in labor.

**Table 1 medicina-58-00087-t001:** Basic characteristics and obstetric history of the study group.

Variable	All	2010	2020
*n* (%)	2208	1175	1033
Mean age, years (SD)	29.9 (±8.1)	28.0 (±8.8)	32.0 (±6.7)
Place of habitation:			
cities >100.000	1276	663 (56.23%)	613 (57.08%)
cities 50.000–100.000	259	139 (11.79%)	120 (11.17%)
cities <50.000	312	166 (14.08%)	146 (13.59%)
village	406	211 (17.90%)	195 (18.16%)
Education:			
primary	72	53 (4.50%)	19 (1.77%)
secondary	708	425 (36.05%)	283 (26.37%)
higher	1266	583 (49.45%)	683 (63.65%)
medical	206	118 (10.01%)	88 (8.20%)
Socioeconomical status:			
low	88	73 (6.27%)	15 (1.40%)
medium	1736	944 (81.03%)	792 (73.95%)
high	412	148(12.70%)	264(24.65%)
Comorbidities:			
none	1553	828 (75.27%)	725 (71.15%)
1	478	222(20.18%)	256 (25.12%)
2	73	39 (3.55%)	34 (3.34%)
3	15	11 (1.00%)	4 (0.39%)
Ongoing pregnancy			
yes	312	215 (18.25%)	97 (9.03%)
no	1940	963 (81.75%)	977 (90.97%)
History of pregnancy			
yes	1436	607 (51.44%)	829 (77.12%)
no	819	573 (48.56%)	246 (22.88%)
History of miscarriage			
yes	393	178 (15.15%)	215 (20.81%)
no	1815	997 (84.85%)	818 (79.19%)
History of delivery			
none	861	595 (52.89%)	266 (24.72%)
vaginal delivery	725	351 (31.2%)	374 (34.76%)
cesarean section	615	179 (15.91%)	436 (40.52%)

**Table 2 medicina-58-00087-t002:** Women’s opinions and preference for intrapartum analgesia.

Variable	2010	2020	*p*
Analgesia in labor			
yes	67.9%	73.9%	0.005
no	15.9%	12.1%	
Choice of analgesia method before the delivery			
yes	92.28%	94.86%	0.047
no	2.74%	1.78%	
Optimum moment of analgesia administration			
at the hospital admission	1.7%	1.2%	0.18
on the onset of pain	9.8%	12.8%	
on the onset of expulsive pains	5.3%	5.6%	
at 3 cm of dilatation	17.4%	14.5%	
at 8 cm dilatation	3.7%	4.3%	
at the moment chosen by the patient	40.2%	39.3%	

## Data Availability

The data presented in this study are available on request from the corresponding author. The data are not publicly available.

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
