# Peer review of "Intrapartum Analgesia—Have Women’s Preferences Changed over the Last Decade?"

_medicina, 2022, doi:10.3390/medicina58010087_

Round 1
Reviewer 1 Report
Dear authors, congratulations on the manuscript. However, before any decision can be made, there are some issues that need attention:
ABSTRACT
You need to define your abbreviations before using them: I do not know what EDA stands for.
INTRODUCTION
You claim that over the past decade, women have been accessing more content on labor through social media. I understand that this is true, but we also know that several of the profiles that we have access to do not comply with scientific evidence (and with that, we have all the antivaxx movement to make my point). As such, why do you believe that having more access to information through social media would imply that women would be better informed, and, as such, change their beliefs on labor pain and methods to relieve it?
MATERIALS AND METHODS
You designed an anonymous survey to be taken by women all over Poland, I suppose. You list several methods to relieve labor pain, but were they explained to the women when they took the survey? For instance, did you explain what pudendal blockade was? Did you explain the difference between acupressure and acupuncture to them?
The inclusion criteria are not disclaimed, and even though they seem pretty obvious, some issues have come to my mind: have you collected data on their gynecological and obstetric history? If so, what information did you collect? What were the issues that you analyzed? For example: were primiparous and multiparous women included? Were they pregnant at the time they were interviewed? Did you include women with any kind of chronic pain condition (such as fibromyalgia or migraine)? Did you consider those issues as confounders? Also, did you collect the kind of social media they mostly accessed? Did you ask them if, by any chance, they were aware of what is scientific evidence? Please, add this information on this section.
RESULTS
One issue that caught my attention is that almost all your sample answered they had a high socioeconomic status. How representative is this regarding the Polish population? By the way, what did you consider as high socioeconomic status? How many minimum wages did you consider?
Also, if possible, turn your results into one or two tables (preferably, one of them should be of participants characteristics, so that we can understand about whom you are talking). All those percentages turn the results a little confusing to read and understand.
DISCUSSION
You mention Swedish studies that have results different from yours, and I believe that you could add what is different regarding pregnancy care in both countries. For instance, do the Swedish public policies make women more secure on labor and more aware on labor pain than does the Polish public policies? What was the socioeconomic status of those Swedish women?
You need to address the issue of the economic status. Usually, there is a tendency of richer women having more access to more expensive analgesic methods, because, well, they can pay for it. What would be the results if this survey was taken in a low-income country where women do not have easy access to any of those methods? What if the study was taken in the USA, with the opioid crisis over there? Do you believe that women would have a different perception on labor pain relief? Also, obstetricians can have tendencies of inducing women to have c-sections and use pharmacological pain relief methods; as such, you need to know if your participants had a OB-GYN or a midwife as counselor.
Even though you comment on those issues in the limitations section, I believe that they need to be addressed in the discussion, and studies from other cultures should be added for comparison.
Author Response
Dear Reviewer 1,
Thank you for your analysis of our manuscript.
We corrected the mistake and removed the abbreviation EDA from the abstract section, changing it to full-text version.
We absolutely agree that wider access to virtual information may be not only beneficial, but also harmful. We simply assumed that social media would give women more knowledge about different methods of analgesia and encourage them to ask their health professionals. Unfortunately, our main problem is short time dedicated to pregnancy appointments in the public health service and the resulting minimization of information provided for the patients.
Different methods of analgesia were not explained in our survey. We took an assumption that our survey would evaluate the level of knowledge and opinions about different methods of analgesia and giving additional information would have influenced the answers, which of course might have been wrong, but unfortunately is not possible to be corrected.
The questionnaire included questions about the women’s medical history, parity, history of cesarean and vaginal deliveries and miscarriages. We included nulliparous, primiparous and multiparous women, some of the women were pregnant at the moment of filling in the survey. We added a table presenting the characteristics of the group.
We asked them about chronic conditions, however without distinction of chronic pain conditions.
Unfortunately, we are not able to recollect what social media they mostly accessed and we did not ask them about their awareness of scientific evidence.
We did not add the wages intervals and depended on women’s self-evaluation of their socioeconomic status. Polish people have a tendency to present themselves as of better socioeconomic status that it is in the reality, and it might have influenced the answers but we have no fundamentals to discredit the respondents opinions. Additionally, women from the lowest income groups had limited accessibility to internet and social media as we mentioned in the limitations section of the manuscript.
Following your remark, we added two tables, one with the basis characteristics of the group (and the previously mentioned medical and obstetric history) and the other with the summary of the results.
We added information about differences in pregnancy care between Sweden and Poland, including the role of midwives. However, we did not attempt to compare the socioeconomic status, as in our study the information was probably less credible than in Sweden. We addressed the issue of the paid epidural analgesia in 2010 and the possible influence of the level of socioeconomic status declared by the participants.
The opioid crisis in the United States definitely has an influence on women’s decisions about pain relief in labor, however we were not able to find direct present studies on the subject.
According to your suggestion, we added to the discussion analysis of studies from low-income countries and other cultures.
In our country, only obstetricians take care of pregnant women, no matter if pregnancy is low- or high-risk. The role of midwives as counselors is limited only to labor preparation courses. Following your remark, we included this information to the discussion section of the manuscript.
We added the information about the socioeconomic status, previously mentioned only in the limitation section, to the discussion section of the manuscript.
Reviewer 2 Report
Dear authors,
Thank you for submitting your work on this very interesting subject that is certainly in favor of young women in labor. It surely expresses the need for more information over this topic to obstetricians, midwives as well as pregnant women.
I have added some comments, suggestions and major concerns to improve the quality of your work.
Wish you luck in re-submitting process
Bets regards
Reviewer
Page 1. Abstract
In this text the authors used for the first time the abbreviation of EDA. To increase the readability of the text, this abbreviation should be written completely here or explained otherwise. Please modify.
The method section
Was this study registered in an European study trial ? please provide the necessary information on the registration identification.
I understand from this report that the inclusion criteria for participating in this study were provided at the social media. Is this correct? Who did inform the patients then, the internet ? Who did check the eligibility of the participants. Please provide information on these matters.
This manuscript conducts and analyzes the results of two surveys at two different time points. When did this take place for the data from 2010 ? in 2010 or 2020? If the data from 2010 were retrieved in 2020, there exists a large bias in accuracy of these data. Are the results from 2010 published in a medical journal? In this case, they should be noted as a reference.
The results section
The authors may consider to rewrite the results section. Some information on percentages of women that are studied, may be modified in order to understand them better. This is specially the case when the authors compare the findings on these women in 2010 with those from 2020. The wording of “respectively” in this regard is sometimes difficult to differentiate.
Did the authors question the participating women about the mode of analgesics that they had well received during the process of labor and childbirth in their personal experience ?
Who did provide the information over the advantages and disadvantages of all pain relief modalities to the women prior to and during the labor?
The authors may also consider to demonstrate the results in one table or two in order to make them more visible.
The authors quite often refer to the results that they have analyzed and retrieved from 2010, the accuracy of which is uncertain. Such study was also not published to refer to for the comparison of results.
The discussion section
The discussion section should be largely rewritten when comparing the results from other studies to make them more sensible and relevant to the study population that is analyzed here.
When discussing the results from a different study project, these may not be applicable to the study population from this authors’ study project. The readers may have the impression that the results from an another study (The Norwegian study for instance) are used to extrapolate to the population from this study. This should be clarified. Please modify.
Conclusion: the conclusions should be relevant to the patients and should have been led to a concrete plan toward the patients. This important information should be highlighted. The conclusion that is driven from this study in 2020 is exact the same as in 2010 and this in spite of significant advancements in medicine and pain therapy. Please elaborate on this issue.
Figure 1.
Is it difficult to understand whether the findings in this figure from Poland are related to the data from the study in 2010 or 2020? Please elaborate on this.
Author Response
Dear Reviewer 2,
Thank you for your analysis of our manuscript.
We corrected the mistake and removed the abbreviation EDA from the abstract section, changing it to full-text version.
The study was not registered as European study trial, it was approved by the Warsaw Medical University Ethics Committee.
The only inclusion criteria were majority (18 years old or more) and female sex. The information about the survey itself and the purpose of the study were presented at the beginning of the survey to be accepted by the participants. The age of participants was verified by one of the questions included in the survey, as to the sex of the participants we had no reason to suspect dishonesty in this matter.
Data from 2010 was collected in 2010 and has not been published before 2021, therefore there is no existing reference and recollection bias.
We modified the results section to make it more understandable, removing “respectively’, and clarifying the presentation of the results.
Women were not asked about their previous personal experience with labor analgesia.
The purpose of the survey was only to collect women’s opinions about the different methods of analgesia, not to inform them about the existing possibilities. Pregnant women in Poland are informed about the possible methods of intrapartum analgesia and their advantages and disadvantages by their obstetricians and additionally in labor by the midwives.
The results from 2010 were obtained in 2010 and not published before to present them now in comparison to results from 2020.
According to your remark, we added two tables, one with the basis characteristics of the and the other with the summary of the results.
We changed the discussion section of the paragraph, including the comparison of different country pregnancy care policies that might influence the choice of labor analgesia and therefore making the comparison with different studies more relevant. We also added analysis of studies from low-income and other culture countries.
Following your suggestion, we corrected the conclusion part of the manuscript, highlighting the existing problem of lack of adequate information about different methods of intrapartum analgesia provided by the health professionals and possible ways of solving it.
Figure 1 includes data from all our participants, both from 2010 and 2021 combined.
Round 2
Reviewer 2 Report
Dear authors,
Thank you very much for providing this revised version and referring to all my questions.
I have no further comments on this revised article.
Best regards
Reviewer